# Autoregressive Action Sequence Learning for Robotic Manipulation

## Abstract

Autoregressive models have demonstrated remarkable success in natural language processing. In this work, we design a simple yet effective autoregressive architecture for robotic manipulation tasks. We propose the Chunking Causal Transformer (CCT), which extends the next-single-token prediction of causal transformers to support multi-token prediction in a single pass. Further, we design a novel attention interleaving strategy that allows CCT to be trained efficiently with teacher-forcing. Based on CCT, we propose the Autoregressive Policy (ARP) model, which learns to generate action sequences autoregressively. We find that action sequence learning enables better leverage of the underlying causal relationships in robotic tasks. We evaluate ARP across diverse robotic manipulation environments, including Push-T, ALOHA, and RLBench, and show that it outperforms the state-of-the-art methods in all tested environments, while being more efficient in computation and parameter sizes. Video demonstrations, our source code and the models of ARP are all included in the supplementary material.

## 1 Introduction

Autoregressive models are the basis of recent breakthroughs in natural language processing (Min et al., 2023). These models predict the next token in a given sequence based on the previous tokens. Autoregressive models are implemented with causal transformers, where each token attends only to preceding ones, and they are trained with the single objective of maximizing the conditional likelihood of each token. Despite their simplicity, autoregressive models such as GPTs (Mann et al., 2020) are shown to demonstrate a reasoning ability that can capture causal dependencies (Prystawski et al., 2024). In this work, we explore the design of a simple autoregressive architecture that can be used for various robot manipulation tasks in diverse environments.

Decision Transformer (DT) and Trajectory Transformer (TT) are the pioneering approaches that use autoregressive models to solve control tasks (Chen et al., 2021; Janner et al., 2021). These methods learn to generate trajectories as $(R_1, s_1, a_1, \ldots, R_T, s_T, a_T)$, where $R_t, s_t, a_t$ respectively denote the reward-to-go (Tamar et al., 2016), the state, and the action at time-step $t$. In contrast, we propose to predict only the future action sequence, and condition the prediction on the current state (or observation). Action sequence learning is more attainable as an objective in robotics applications, where the underlying reward functions are unknown or ill-defined, and observations, given as images or point clouds, are high-dimensional and cannot be predicted accurately (Kroemer et al., 2021). DT and TT are mainly used in tasks where low-dimensional state variables are fully observed. To deal with uncertainty, our model generates a new action sequence after every $k$ time-steps, using an updated new observation and following the Model Predictive Control (MPC) approach. Action sequence modeling also enables a better leverage of the underlying causal dependencies in robotic tasks. Examples of such causal dependencies include: logical dependencies, where low-level actions depend on high-level plans, spatial dependencies, where orientation depends on the end-effector's position, and temporal dependencies, where latter actions depend on earlier ones. We showcase the action sequence designs of our real robot experiment, Push-T, ALOHA, and RLBench in Figure 3.

We propose the Chunking Causal Transformer (CCT), an auto-regressive model that is tailored for robotic tasks. CCT extends the next-token predictions of causal transformer to support multi-token predictions in a single pass. CCT predicts the future tokens, or a chunk of actions, from empty tokens rather than from the original sequence, as illustrated in Figure A1. In doing so, CCT is more

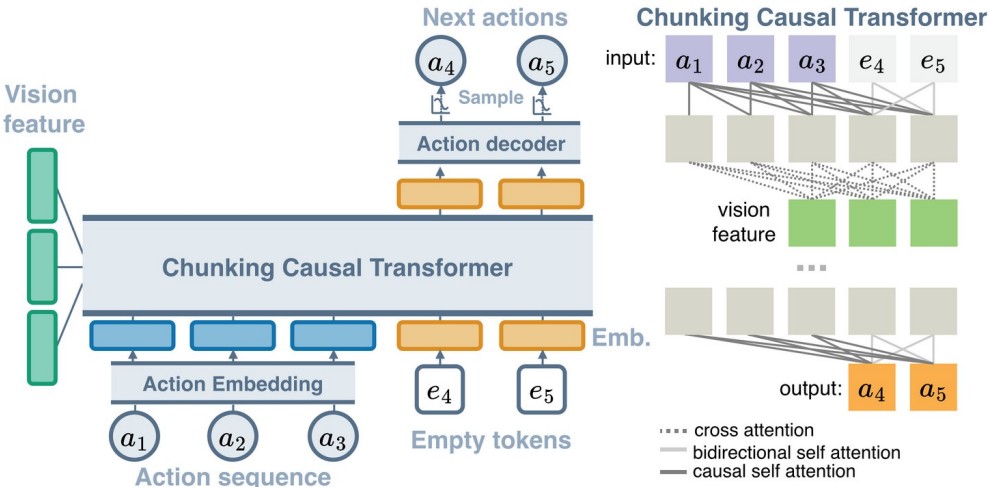

Figure 1: **Autoregressive Policy Architecture.** A sequence of past actions and a chunk of empty tokens are concatenated and projected into embeddings. The empty tokens correspond to future actions, which are unknown and need to be predicted. These embeddings are fed into our Chunking Causal Transformer (CCT) along with the vision features of the current observation. CCT alternates between self-attention within the input embeddings and cross-attention with the vision features. Self-attention is causal for the known input actions and bidirectional among the empty tokens. Distributions of future actions are decoded from the updated embeddings of the empty tokens.

efficient by requiring fewer inference passes and delivers stronger performance because grouping actions in chunks is critical for robotic tasks that require high control frequency (Zhao et al., 2023). Further, we design a novel attention interleaving strategy that allows CCT to be trained efficiently with teacher-forcing, as shown in Figure 4. While action chunking has been previously introduced in the Action Chunking Transformer (ACT) by Zhao et al. (2023), ACT is a one-step prediction model. Instead, we employ action chunking for auto-regressive models. Our ablation studies show that both auto-regression and action chunking are the key factors behind the strong performance of the proposed model, as shown in Table 1 and Figure 7.

To summarize, our contributions are threefold. (1) We propose the Chunking Causal Transformer (CCT), which extends the next-token prediction of causal transformer to multi-token prediction for auto-regressive models. We also design a novel attention interleaving strategy that allows CCT to be trained efficiently with teacher-forcing. (2) Based on our CCT, we present the Auto-regressive Policy (ARP), a model that learns to generate action sequences auto-regressively for solving robotic manipulation tasks. The ARP architecture is summarized in Figure 1. (3) We evaluate ARP across Push-T (Chi et al., 2023), ALOHA (Zhao et al., 2023), and RLBench (James et al., 2020), three environments with diverse manipulation tasks, as outlined in Figure 2. Our study shows that ARP outperforms all environment-specific SoTAs, while being more efficient computationally and using smaller parameter sizes, as summarized in Figure 5. In addition, we evaluate ARP with a real robot on a challenging, contact-rich nut-tightening task, as shown in Figure 8.

## 2 RELATED WORK

**Learning robotic manipulation from demonstrations.** Imitation learning enables robots to learn to perform tasks demonstrated by experts (Zare et al., 2024). Recently, various methods have been developed for manipulation learning with different task constraints and control modalities. Notably, Chi et al. (2023) proposed the Diffusion Policy (DP) method for solving the Push-T task. Zhao et al. (2023) proposed the Action Chunking Transformer (ACT) for bi-manual manipulation tasks in the ALOHA environment. Goyal et al. (2024) proposed RVT-2 for language-conditioned tasks in the RLBench environment (James et al., 2020). We outline these environments and the corresponding state-of-the-art (SoTA) solutions in Figure 2 and Figure 6, respectively. In contrast, our proposed

**Push-T**     **ALOHA**     **RLBench**

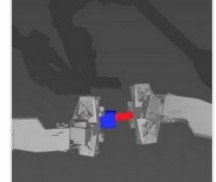 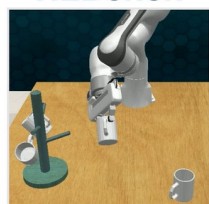

*"Push the T-shape object to closely overlap the green region*

- Action space: 2d pointer coordinate
- Need spatial reasoning
- Long-horizon multi-modality

*"Pick up and insert the stick into the hub with two arms."*

- Action space: 14 joint positions
- High control frequency
- Very short horizon
- Bimanual manipulation

*"Place 3 cups onto the cup trees."*

- Action space: 6DoF end-effector pose and binary gripper states
- Low control frequency with waypoints
- Diverse tasks with language condition

Figure 2: **Overview of the simulation environments.** We evaluate our technique on Push-T, ALOHA, and RLBench, three task suites with significantly different properties and requirements. Push-T (Chi et al., 2023) is a task that requires many steps to complete (long horizon) and where the same sub-goals can be reached in various ways (multi-modality). ALOHA (Zhao et al., 2023) has a high-dimensional action space (14 joints of two robot arms), a high control frequency (50Hz), and a short time limit (8 seconds). RLBench (James et al., 2020) has only the gripper pose as an action but contains 18 different language-conditioned tasks.

auto-regressive policy is a universal architecture that outperforms each environment-specific SoTA on Push-T, ALOHA, and RLBench.

**Sequence models for control tasks.** In addition to the Decision Transformer (DT) and the Trajectory Transformer (TT), recent works such as OpenVLA (Kim et al., 2024) and ManipLLM (Li et al., 2024) have looked into fine-tuning a large language model (LLM) such as LLaMA (Li et al., 2024) to directly include target end-effector poses within text-based responses. Despite their impressive results, these approaches are limited to low-frequency control tasks that rely on end-effector waypoints (Kim et al., 2024). Moreover, the reliance on resource-intensive LLMs leads to large computational overhead, even for tasks that could be addressed with lightweight models. Without these constraints, our auto-regressive policy outperforms SoTAs in multiple environments while being more efficient in MACs (number of multiply-accumulate operations) and parameter sizes.

**Hierarchical policies.** Planning actions on multiple levels of abstraction is an important ability (Pateria et al., 2021). Existing methods generally separate the designs of low-level and high-level policies, and uses different modules for the different levels of abstraction (Le et al., 2018; Pateria et al., 2021; Belkhale et al., 2023; 2024; Chen et al., 2024b). This complicated procedure prohibits a wider application. In contrast, our auto-regressive policy predicts a sequence of actions of different levels of abstraction by using a single model.

## 3 METHOD

In this section, we present the Auto-regressive Policy (ARP), which predicts actions using the Chunking Causal Transformer (CCT). The architecture is summarized in Figure 1.

**Action sequence design.** Unlike natural language, robot actions lack a universal vocabulary. As shown in Figure 2, different robot tasks may require drastically different types of actions. We therefore propose to represent actions as structured sequences that follow a format that is pre-defined for each family of tasks. Figure 3 showcases the formats of the action sequences generated in our real robot experiment, Push-T, ALOHA, and RLBench tasks.

**Action embedding and decoding.** Language models map each word to a continuous vector called word embedding. The word embeddings of the input sentences are fed into a causal transformer. The distribution of the next word is decoded from the output embedding of the last word with a linear layer. Figure A2 and A3 illustrate our embedding and decoding methods for robot actions.

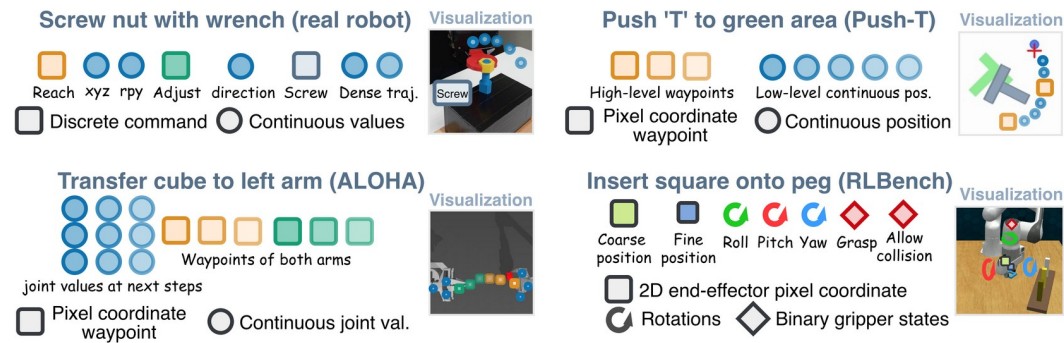

Figure 3: **Learned Action Sequences.** In Push-T, our model predicts a sequence of high-level waypoints, followed by a sequence of low-level positions that connect the waypoints together and form in the pushing trajectory, analogous to hierarchical planning (Hafner et al., 2022). In ALOHA, we predict the joint values and then the end-effector waypoints conditioned on the joint values, a process akin to forward kinematics (Kucuk & Bingul, 2006). We bypass the waypoint generation during inference. In RLBench, we predict the target end-effector's position first, then the gripper rotation and state in that position. For our real robot experiment, we define a set of primitive actions, as detailed in section 4.3. We predict the action type and then the continuous values of that action.

Discrete actions are embedded by a table lookup on a weight matrix and decoded into a categorical distribution with a linear layer, similar to words in language modeling. Continuous actions are embedded with a linear layer and decoded into the parameters of a Gaussian mixture distribution with another linear layer. Actions that are defined as pixel coordinates are embedded by retrieving the point-wise features at the coordinates on a visual feature map. The output spatial distribution is obtained by multiplying the output embedding with the visual feature map, and converting the result into a 2-d heatmap with the up-sampling operator from RAFT (Teed & Deng, 2020).

**Chunking causal transformer.** Figure A1 illustrates the essential difference between a causal transformer and our CCT. A causal transformer modifies the token embedding with causal attention so that the last token becomes the next token. Our CCT modifies the token embedding with causal attention for the known tokens $a_i$ (past actions) and bidirectional attention for the empty tokens $e_i$ (future actions). The empty tokens become the next tokens. This allows the prediction of multiple next tokens at once in a single forward pass by adding empty tokens. The advantages are two-fold: (1) A better accuracy is achieved because error accumulation is reduced when actions are grouped in chunks and executed as one unit (Zhao et al., 2023). (2) A better efficiency is achieved with fewer forward passes because each forward pass predicts multiple tokens at once. We study the impacts of action chunking in detail in Section 4. In ARP, CCT alternates between self-attention within the input embeddings and cross-attention with vision features, as in Figure 1. We extract vision features from a standard backbone identical to the ones used in SoTA methods, as detailed in section 4.

**Train-time attention interleaving.** During training, a causal transformer is taught to predict each token in a given sequence by consuming all previous ground-truth tokens as input. This training strategy is named teacher-forcing (Williams & Zipser, 1989). As shown in Figure 4, only a single forward pass is required for training samples such as $a_1, a_2, a_3 \rightarrow a_4$ (predict $a_4$ from $a_1, a_2, a_3$), $a_1, a_2 \rightarrow a_3$, and $a_1 \rightarrow a_2$. Causal transformers are therefore efficiently trained with teacher-forcing. We follow this teacher-forcing strategy. However, training CCT yields separate forward passes per chunk. For example, the prediction of $a_4$ depends on $a_2, a_3$, as in $a_1, a_2, a_3, e_4 \rightarrow a_4$, but $a_2, a_3$ need to be replaced with $e_2, e_3$ to predict them from $a_1$, as in $a_1, e_2, e_3 \rightarrow a_2, a_3$. This prohibits the use of a single forward pass for both $a_1, a_2, a_3, e_4 \rightarrow a_4$ and $a_1, e_2, e_3 \rightarrow a_2, a_3$. Note $a_i$ denotes the $i$-th action and $e_i$ denotes the empty token for $i$-th action. This issue increases the training cost and drastically complicates the training procedure.

To resolve this, we first divide the attention into three categories: (1) causal attention within the known tokens, (2) bidirectional attention within the empty tokens, and (3) causal attention between the empty and the known tokens, as marked in Figure 4. Figure A4 shows that the causal attention within known tokens is computed repeatedly. We avoid this redundancy by precomputing the causal

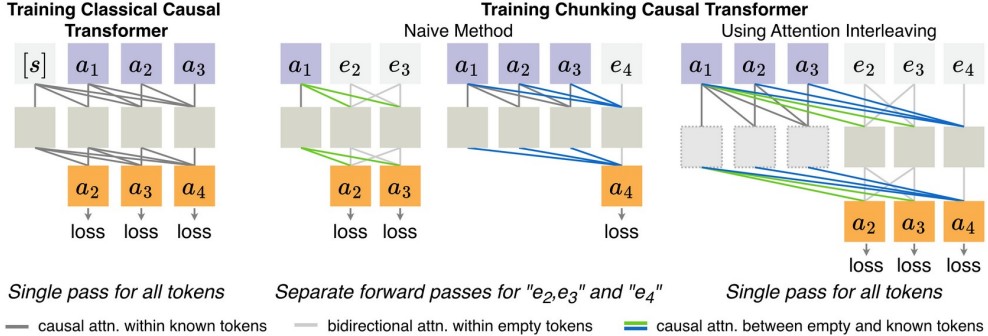

Figure 4: **Training Chunking Causal Transformer (CCT) with Teacher-forcing.** Causal transformers are trained efficiently with only a single forward pass for all tokens in a given sequence. However, suppose $a_2, a_3$ and $a_4$ are in separate chunks, the CCT forward passes of predicting $a_2, a_3$ and $a_4$ cannot be merged directly. Naively running separate passes significantly increases computation costs, as in Figure A4. With the proposed attention interleaving, we precompute and cache the causal attention results for all known tokens $a_1, a_2, a_3$. For the empty tokens $e_2, e_3, e_4$, we combine their bidirectional attention (lightgray) and the causal attention towards known tokens (green, blue). Since all the computations related to the known tokens are cached, we can update all the empty tokens in a single pass, regardless of the number of tokens. An example of attention interleaving is provided in the `Video/attention-interleaving-tour.mp4` in the supplementary.

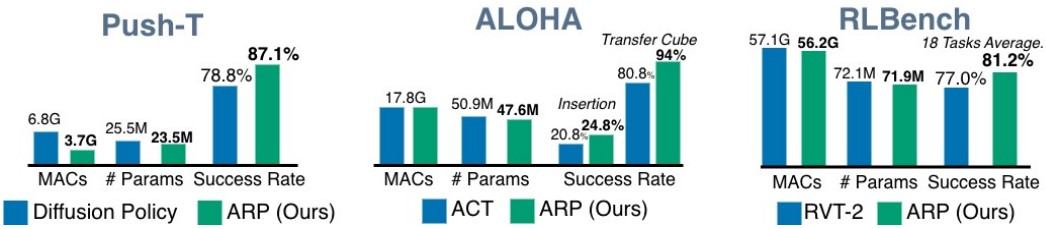

Figure 5: **Comparing our proposed Autoregressive Policy to the SoTA of each environment.** Our autoregressive policy (ARP) outperforms environment-specific SoTA and is more efficient in MACs (number of multiply-accumulate operations) and parameter sizes. We report the results of the transformer version of the diffusion policy because of its overall better performance. The RVT-2 (Goyal et al., 2024) results are obtained without including timesteps in the input, similarly to ARP. All MACs and parameter sizes are measured using THOP (Zhu).

attentions for all known tokens and caching the results. For the empty tokens, we combine their inner bidirectional attention and the causal attention toward cached known tokens. This enables a single forward pass of all tokens in three attention operations, regardless of the number of tokens. We name this procedure *attention interleaving*. Figure A4 demonstrates the reduced MACs of training with attention interleaving. We implement attention interleaving as an internal acceleration mechanism of the transformer layer, which is transparent to other network modules. Note that attention interleaving is only used during training and incurs no additional inference cost.

**Inference.** During the test rollouts, we extract vision tokens from the current observation and provide them as input to ARP, which then generates actions auto-regressively by sampling from the decoded action distribution and appending the selected actions to the existing action sequence. This process of generating and executing actions is repeated until episode termination (success, failure, or reaching the time limit). Actions are generated according to the sequence formats shown in Figure 3 with pre-defined chunk sizes and target lengths per task. We provide additional implementation details and hyper-parameter values in Section 4 and Appendix A.2.

## 4 EXPERIMENTS

In this section, we investigate how the Auto-regressive Policy (ARP) performs compared to the existing methods that were designed specifically for each environment. In addition, we examine

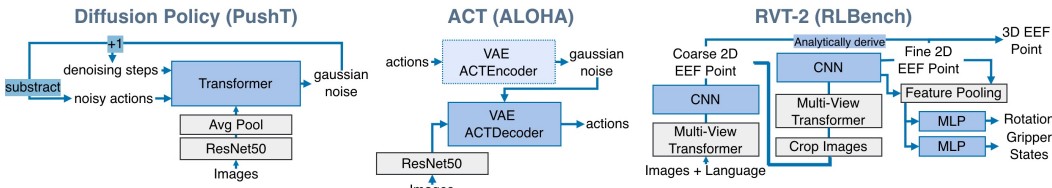

Figure 6: **Overview of SoTA solutions on Push-T, ALOHA, and RLBench.** Diffusion Policy (DP) (Chi et al., 2023) iteratively subtracts Gaussian noises from noisy actions. The transformer network predicts the Guassian noise at each step. Action Chunking Transformer (ACT) (Zhao et al., 2023) is a VAE architecture that predicts actions directly from images and Gaussian noises. RVT-2 (Goyal et al., 2024) is a hybrid and more complex model, but it is trained directly with behavior cloning and it does not require a generative framework such as diffusion or VAE.

whether auto-regression and action chunking are the primary contributors to the performance gains and evaluate how well existing methods perform across different environments. Further, we verify ARP on a challenging nut-tightening task with a real robot. Finally, we demonstrate that ARP can estimate the likelihood of robot actions and predict actions based on human inputs. All of our source code and the pre-trained models are included in the supplementary material and will be publicly released upon acceptance. A single-file implementation of our auto-regressive policy can be found at `Code/arp.py` in the supplementary material.

## 4.1 COMPARISON WITH STATE-OF-THE-ART

**Setup.** We compare the auto-regressive policy (ARP) against the SoTA solutions in Push-T, ALOHA, and RLBench environments. Push-T is a single task. ALOHA consists of two tasks: insertion and cube transfer. RLBench includes 18 tasks, each with multiple language variants. These environments are illustrated in Figure 2 and Figure A5. For Push-T and ALOHA, we train a separate policy for each task. For RLBench, a single policy is trained for all 18 tasks. In Push-T, the policy observes the last two $96 \times 96$ RGB frames, and predicts a window of future 2-d pointer positions as actions. In ALOHA, the policy observes the current $480 \times 640$ RGB frame and predicts a window of future 14-dimensional joint positions. In RLBench, the policy observes four RGBD $128 \times 128$ images and predicts the next target end-effector pose and gripper states. Existing SoTA techniques in these environments are outlined in Figure 6. We use the same vision backbones as the SoTA solutions to extract vision tokens, namely ResNet50 (He et al., 2016) for Push-T and ALOHA, and Multi-View Transformer (Goyal et al., 2023) for RLBench. We use the same training data, number of episodes, optimizer configuration, and evaluation frequency as the SoTA solutions. We detail the full list of hyper-parameters, such as the number of layers, sequence lengths, chunk sizes, and optimizer setups in Appendix A.2. Success rates for Push-T and RLBench are averaged over three independent runs. ALOHA's results are averaged over five runs.

**Results.** Figure 5 shows that our auto-regressive policy (ARP) outperforms environment-specific SoTAs while being more computationally efficient. Table A4 compares the per-task success rates of our ARP and RVT-2 (Goyal et al., 2024). In addition, we report the result of ARP[+], which shares the same network definition with ARP but has more layers. The MACs / parameter sizes of RVT-2, ARP, ARP[+] are 72.1M/57.1G, 71.9M/56.2G, and 73.8M/57.4G, respectively. Notably, ARP[+] achieves an average success rate of 86% with a minor increase in computational cost. Note that the success rate of RVT-2 was originally reported as 81.6%. But this was achieved by using a logical time-step as an extra input, which indicates task progress (*sub-task milestone*). Many existing works have followed this convention (Shridhar et al., 2023; Goyal et al., 2023). However, this information is unavailable in real applications. Thus, we train all our RLBench models without the input time-step.

## 4.2 ANALYSIS

**Does the performance gain come from auto-regression?** Our action sequence design incorporates additional inputs for Push-T and ALOHA, as shown in Figure 3. These inputs are automatically extracted from the demonstration trajectories. In Push-T, the high-level waypoints are simply uniformly sampled down from the low-level trajectories and then discretized. In ALOHA, the pixel coordinates of the end-effector are computed from the joint values with the robot's forward kine-

Table 1: **Auto Regression versus One-step Prediction and SoTA.** The baseline refers to the SoTA (Diffusion Policy for Push-T and ACT for ALOHA). Auto Regression is our proposed approach, where actions are generated auto-regressively. One-step prediction shares the same implementation and training data as the proposed approach but generates the entire action sequence in a single step.

| Generation Mode | Push-T | ALOHA | |
| --- | --- | --- | --- |
| | | Cube Transfer | Insertion |
| Baseline | 78.8 | 80.8 | 20.8 |
| Auto Regression | **87.1** | **94** | **24.8** |
| One-step Prediction | 77.6 | 81.2 | 21.2 |

Table 2: **Evaluation of existing methods on various environments.** ACT, a VAE-based method, performs competitively across all environments, whereas Diffusion Policy struggles in ALOHA and RLBench. While we believe stronger diffusion-based architectures can be developed in the future, our results suggest that simpler architectures tend to be more robust across diverse tasks.

| Method | PushT | ALOHA | | RLBench |
| --- | --- | --- | --- | --- |
| | | Cube Transfer | Insertion | |
| Diffusion Policy | 78.8 | 10 | 1.6 | - |
| ACT | 77.5 | 80.8 | 20.8 | 69.8 |
| ARP (Ours) | **87.1** | **94** | **24.8** | **81.2** |

matics and the camera parameters. It is possible that the performance gain of ARP originates from this extra information instead of our proposed auto-regression architecture.

Table 1 compares the success rates of auto-regression and one-step prediction in Push-T and ALOHA. Both share the same implementation, with one-step prediction generating the entire sequence at once by setting the CCT chunk size to the full sequence length. The baseline refers to the diffusion policy for Push-T and ACT for ALOHA. The results clearly show that auto-regression is the key factor behind the better performance. Our intuition can be explained through an example: imagine task $B$ is difficult, but solving task $A$ first, followed by solving task $B|A$ (task $B$ given the result of task $A$), is much easier. An auto-regressive model follows this sequential process, solving task $A$ first and then leveraging the result to make task $B$ more feasible. In contrast, a one-step model attempts to predict both tasks simultaneously, treating $A$ and $B$ as independent problems. While the one-step model may solve task $A$ implicitly as part of solving task $B$, it does not explicitly take advantage of the problem structure and is therefore prone to shortcuts. This phenomenon has been explored in more depth for NLP tasks by Prystawski et al. (2024).

**Do existing methods work in different environments?** Table 2 shows how existing methods perform in different environments. When testing in a new environment, we keep the same architecture but adapt the vision backbone and optimizer to the environment's established setup. RVT-2 was not implemented for Push-T and ALOHA, as it is designed for sparse waypoint predictions, which are incompatible with the high-frequency actions required in these tasks. We did not implement the diffusion policy for RLBench, as it refines actions from gaussian noise, which conflicts with the common practice in RLBench of predicting actions in discrete spaces. While 3D Diffuser Actor (Ke et al., 2024) reports competitive results on RLBench, it uses a completely different architecture.

Table 2 reveals that ACT, a VAE-based method, performs competitively across all environments, whereas the diffusion policy struggles to deliver meaningful performance in ALOHA. This outcome is surprising, given the recent popularity of diffusion-based techniques. While we believe a strong diffusion architecture, like 3D Diffuser Actor on RLBench, could be developed for ALOHA, this suggests that simpler architectures could be more robust across a wider range of tasks and environments. Our auto-regressive policy is trained with a single objective: to maximize the conditional likelihood of each action in a sequence. We believe this simplicity contributes to its robust performance across diverse environments.

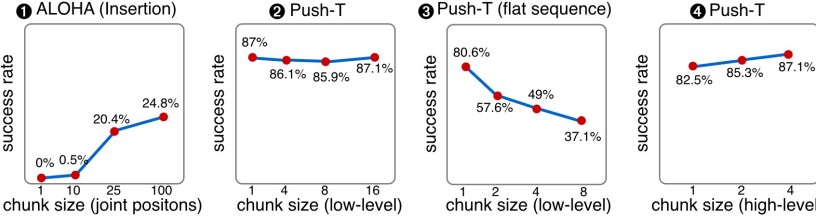

Figure 7: **Impact of chunk size on performance.** Our results suggest that the optimal chunk size depends on both the task and the action sequence design. Therefore, the ability of our chunking causal transformer to flexibly adjust chunk sizes is essential for maximizing performance.

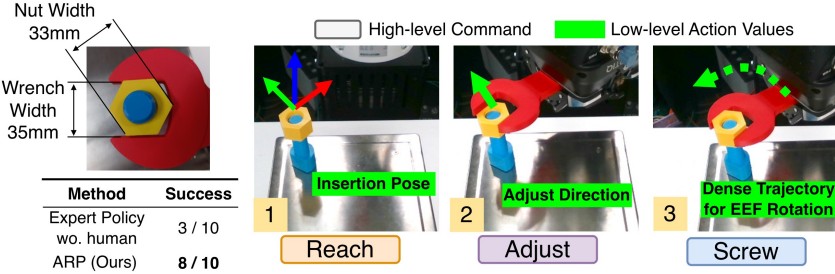

Figure 8: **Real robot experiment.** Our ARP learns to adaptively select high-level commands and generate low-level action values, including position adjustment after unsuccessful insertion. In doing so, we achieve a success rate of 8/10 in this nut-screwing task that requires a precise tool alignment. The bolt's position (blue) and nut's height (yellow) are randomized at every episode.

**Does action chunking improve performance?** Instead of predicting only the next token, our chunking causal transformer (CCT) is able to predict multiple next tokens, that is, a chunk of actions. Figure 7 illustrates the relationship between chunk size and success rate. The first plot shows that larger chunks significantly improve policy performance, a trend also observed by ACT (Zhao et al., 2023). This advantage of chunking actions seems generalizable to high-frequency control in short-horizon tasks. Interestingly, while larger chunk sizes for joint positions improve performance, one-shot predictions, where both end-effector waypoints and joint positions are predicted simultaneously, yield inferior results, as in Table 1.

The second plot indicates that for Push-T, policy performance is largely insensitive to the chunk size of low-level trajectories because the standard deviation of the success rate ranges between 1 and 2. In this case, a moderate chunk size can be a better choice, given the common practice of executing only the first few predicted actions and then rerunning the policy, a test-time technique reduces error accumulation. This technique benefits from a moderate chunk size through early termination of autoregressive generation without sacrificing performance or computational efficiency.

In the third plot, we explore a different action sequence format for Push-T, where we remove high-level waypoints and flatten the trajectories into a vector, as detailed in Figure A6. This design yields a completely different trend, with the policy performing well only when the chunk size is 1. The fourth plot shows that increasing the chunk size for high-level waypoints improves policy performance. These findings demonstrate that the optimal chunk size depends on both the task and the action sequence format. As a result, CCT's ability to flexibly adjust chunk sizes is essential for maximizing performance. Currently, chunk sizes are manually specified for each task, but a more principled approach to adaptively adjusting chunk size would be ideal, as discussed in section 5.

### 4.3 REAL ROBOT EXPERIMENT

**Setup.** We evaluate ARP on a challenging tight nut-screwing task using a real robot, which requires precise alignment between the nut and wrench with a tolerance of 2mm, as shown in Figure 8. In each episode, the bolt (blue) is randomly placed on a $20 \times 20$ cm$^2$ table, while the height of the nut (yellow) is randomized along the 6cm tall bolt. The orientations of both the bolt and nut are also

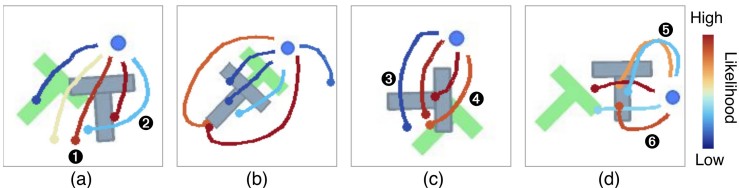

Figure 9: **Trajectory Likelihood Estimation.** ARP generally assigns higher likelihoods to effective trajectories over futile ones, and demonstrates its understanding of action multi-modality as in subfigure (b). The likelihood inference ability of ARP can help identify model weaknesses and find defective demonstrations. All trajectories are human-drawn and are not part of the training set.

randomized per episode. We define three primitive actions: reach, adjust, and bolt. At each step, our ARP predicts a high-level command to select the action and then generates corresponding low-level action values. For example, ARP first predicts the reach command and an insertion pose. Next, the robot attempts to insert the wrench. After every unsuccessful attempt, the policy predicts the adjustment direction to adjust the wrench's position and reattempt insertion. Once the insertion succeeds, the policy switches to the screw command and predicts a dense trajectory to follow in order to rotate the end-effector around the wrench. All commands are automatically predicted by the autonomous model instead of being manually specified. An impedance controller stops unsuccessful insertions based on force feedback. We deploy this model on a Kuka LBR iiwa robot. We use $480 \times 640$ RBG-D observations from a single RealSense D415 camera. We use MVT as the vision backbone. To simplify the task, we assume the wrench is already grasped by the robot in a pre-defined position. An episode is considered successful if a screw action is completed after no more than three attempts to align the wrench on the nut. We trained ARP using 70 demonstrations collected from an expert policy. The expert policy uses Foundation Pose (Örnek et al., 2023) to estimate insertion pose, with human operators providing fine-grained adjustments.

**Results.** Figure 8 shows that ARP screws nuts successfully in 8 out of 10 episodes, while the expert policy only has 3 successes out of 10 without human interventions. Most episodes succeeded without any adjustments because we used the adjusted successful insertion pose as the label for the reach command during training. To test ARP's adaptive adjustment ability, we add a uniform noise ranging from -5mm and 5mm along the normal plane of the insertion pose. Despite the added noise, our ARP still succeeds in 6 out of 10 trials, with an average number of 1.6 adjustments per trial.

## 4.4 QUALITATIVE VISUALIZATION

We showcase all the evaluation tasks in Figure A5. Video demonstrations of ARP in simulation and in the real world are available in the file `Video/demo.mp4` in the supplementary material. In this section, we show two key advantages of ARP: (1) estimating the likelihood of given robot actions, (2) and predicting actions conditioned on human input.

**Likelihood inference.** To generate the next token $a_n$, an auto-regressive model estimates the conditional probability $P(a_n|a_1, ..., a_{n-1})$. Using the product rule, the model can estimate the joint probability $P(a_1, ..., a_n) = \prod_{i=2}^{n} P(a_i|a_1, ..., a_{i-1})P(a_1)$ for any given sequences, a capability that more advanced generative frameworks such as VAE and diffusion lack. Figure 9 shows for different trajectories the likelihood estimated by ARP. All these trajectories are human demonstrations. ARP generally assigns higher likelihoods to effective trajectories and lower likelihoods to futile ones. For instance, in sub-figure (b), ARP assigns high likelihoods to two symmetrical trajectories around the T object, demonstrating its understanding of action multi-modality. However, some likelihood assignments are less intuitive. For example, trajectories ❶, ❹, and ❻ receive moderately high likelihoods, yet they may not bring the T-shape object closer to the green target, at least not better than the low-likelihood trajectories ❷ and ❸. ❺ marks two similar trajectories, yet they have different likelihoods. We believe that this type of likelihood inference can help identify the model's weaknesses and eliminate defective demonstrations from the training data.

**Prediction with human guidance.** Auto-regressive models generate future tokens conditioned on the previous sequence. In Figure 10, we illustrate examples of trajectories of ARP (blue) in Push-T, predicted conditionally on human-drawn initial trajectories (red). The first row (green) shows predictions under correct guidance, where the intention is to complete the task successfully. The

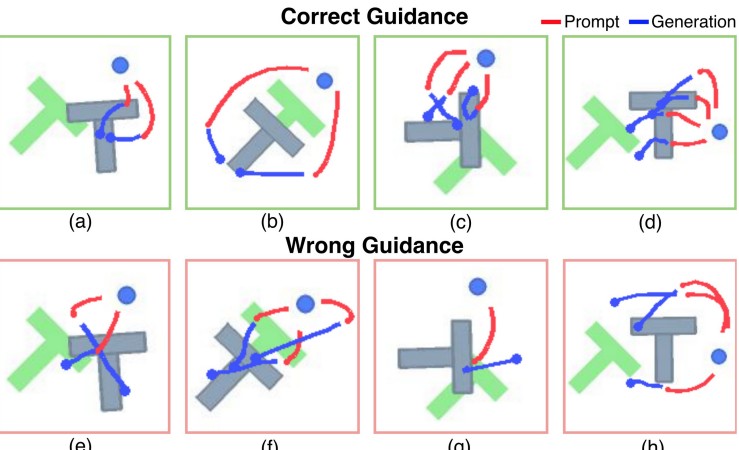

Figure 10: **Trajectory Prediction based on Human Guidance.** We show predicted trajectories of ARP (blue), conditioned on human-drawn trajectories (red). The correct guidance is given with the intention of completing the task, and the wrong guidance is aimed at failing the task. ARP performs as expected under correct guidance. Under wrong guidance, ARP recovers from failure in subfigure (g), avoids further mistakes in subfigure (h), and amplifies the errors in subfigure (e) and (f), which reflects out-of-distribution behavior, as the training set consists only of successful demonstrations.

second row (pink) is based on a wrong guidance with the intention of failing the task. ARP completes the trajectory correctly given a correct initial part. Given a wrong initiation, sub-figure (g) shows ARP's recovery from failure by correcting its initial trajectory. In sub-figures (e) and (f), however, ARP amplifies the initial error by pushing further in the wrong direction. This behavior likely reflects ARP's out-of-distribution response, as the training set consists only of successful demonstrations.

## 5 DISCUSSION

We have shown that ARP is a strong and universal architecture that can be trained to perform diverse manipulation tasks. In the following, we discuss its limitations and potential future directions.

**Learning to plan.** Planning is a key ability of intelligent agents. It requires the agent to reason not only about its actions but also their impacts on its environment (Garrett et al., 2021). Motivated by the reasoning capacity of auto-regressive models in NLP, a promising direction is to incorporate planning into ARP. One possible solution is to predict sequences of both states and actions. States in robotics are typically high-dimensional, such as images or point clouds. Therefore, it would be desirable to predict only key states instead of generating every frame in the future. To solve this problem, ARP can be extended to generate future states by using recent hybrid architectures of auto-regression and diffusion, such as Diffusion Forcing (Chen et al., 2024a),

**Interactive robot learning.** Human-Robot collaboration improves efficiency by allowing the robot to recover from its errors (Mukherjee et al., 2022; Liu et al., 2023). One possible future direction is to integrate active learning techniques into ARP to learn from immediate human feedback. The auto-regressive mechanism naturally supports conditioning action prediction on human input. Moreover, ARP can estimate the likelihood of action sequences. Likelihood is a common measure for identifying the most informative samples in active learning (Taylor et al., 2021). This can be used, for example, to prioritize demonstrations of tasks where the robot encounters more difficulties.

**Adaptive action sequence learning.** Despite ARP's impressive performance, it still requires a manual design of action sequence formats and chunk sizes for each environment. Developing a general approach to automatically determine the optimal chunk size would not only improve ARP's performance, but also deepen our understanding of the action chunking techniques for robot imitation learning in general. We discuss when and why action chunking matters in Appendix A.3. Additionally, unlike natural language, robot actions lack a universal vocabulary. A promising direction is to design a universal robot action language that is applicable across multiple environments, which would reduce the cost of defining new actions, unify training datasets, and improve generalization.

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

# A APPENDIX

## A.1 CODE AND PRETRAINED MODELS

The source code of our auto-regressive policy is included in the supplementary folder `Code`. Please check `Code/README.md` for instructions on installation, dataset setup, and downloading pre-trained models from an anonymous server.

## A.2 HYPER-PARAMETERS AND IMPLEMENTATION DETAILS

In this section, we provide a full list of hyper-parameters in Table A1, Table A2, and Table A3 for Push-T, ALOHA, and RLBench, respectively, along with comments on selected hyper-parameters to provide additional implementation details.

**Model.** The mlp size denotes the hidden feature dimension of the MLP network within the standard multi-head attention operation. The number of latents refers to the number of Gaussians for the Gaussian mixture distribution used to decode continuous actions. The backbone denotes the network used to extract the vision features. We use the ResNet50 for Push-T and ALOHA, and Multi-View Transformer (MVT) (Goyal et al., 2023) for RLBench, identical to the ones used in Diffusion Policy, ACT, and RVT2.

**Action Sequence.** The horizon refers to the number of actions predicted at each step, while the number of action steps indicates how many actions are actually executed, with the remainder discarded. We adopt the same horizon and action steps as state-of-the-art methods. In Push-T, the chunk size for both high- and low-level actions matches the horizon, meaning all high-level points are predicted in one chunk, followed by all low-level points. Yet, interestingly, as shown in Table 1, combining these two chunks into a single-step prediction degrades performance. For RLBench, which uses the next key end-effector pose as the control interface, there is no need for high-frequency actions, so neither the horizon nor action steps apply. Instead, low-level robot movements are generated using RLBench's built-in RRT planner. We use a chunk size of 2 for binary gripper states and a chunk size of 1 for end-effector positions and rotations. For example, ARP first predicts the roll, followed by pitch and yaw of the rotation Euler angle. We follow the strategy of RVT-2 to predict coarse positions and then refine them by zooming into the images (with updated vision features) to obtain more accurate positions. The end-effector positions are predicted in 2-d, and the 3-d positions are derived from the 2-d coordinates of each viewpoint.

**Train& Eval.** The observation $2 \times 96 \times 96 \times 3$ represents 2 frames of RGB images, each with a resolution of 96x96 pixels. For RLBench, the observation $4 \times 128 \times 128 \times 4$ refers to RGBD images (with one depth channel) at 128x128 resolution, captured from 4 cameras. In ALOHA, the maximum evaluation steps of 400 and control frequency of 50Hz indicate an evaluation time limit of 8 seconds. LAMB refers to the large batch optimizer proposed by You et al. (2019). We use the same number of training steps, evaluation frequency, optimizer, learning rate, and learning rate scheduler as used by the SoTA solutions.

## A.3 DISCUSSION ON ACTION CHUNKING

Action chunking has a clear downside – when predicting multiple actions at a time, the agent doesn't receive information about what state was observed after the first action. This means that the agent is operating with less information than if a single-step prediction was used. At the same time, in a MDP the state is guaranteed to be a sufficient statistic for the optimal policy. Given this information, why should action chucking be useful?

We propose two main reasons. First, as has been explored in other imitation learning works, using expert data means that the dataset often lacks information on how to recover from errors, which means that predictions grow worse over time. Using longer action chunks effectively shortens the time horizon. However, we find that action chunking still has noticeable benefits even when the state is well-covered, such as in the Push-T environment. Additionally, this problem becomes less severe as the dataset grows – when the prediction error goes to zero, so does the effect of error recovery Ross et al. (2011).

Table A1: Hyperparameters used in our experiments on Push-T.

| Hyperparameter | Value |
|---|---:|
| *Model* | |
| number of layers | 30 |
| embedding size | 64 |
| mlp size | 256 |
| number of latents (gmm) | 4 |
| backbone | RN50 |
| *Action Sequence* | |
| horizon (low-level) | 16 |
| horizon (high-level) | 4 |
| number of action steps | 8 |
| chunk size (low-level) | 16 |
| chunk size (high-level) | 4 |
| *Train & Eval* | |
| observation | RGB $2 \times 96 \times 96 \times 3$ |
| control frequency | 10 |
| maximum evaluation steps | 300 |
| train epochs | 2000 |
| eval frequency | 50 |
| batch size | 128 |
| learning rate | 0.0001 |
| learning rate scheduler | cosine with restart |
| optimizer | AdamW |

The second and perhaps stronger explanation is that if the demonstrations are non-Markov, the Markov policy that maximizes single-step accuracy is \*not necessarily the optimal policy\*. This is true even even if the demonstration policies are optimal, and even in the limit as data and model capacity become infinite. This is because the state occupancy measure is not convex with respect to the policy, so linear combinations of policies can lead to state distributions that are not linear combinations of the demonstration state distributions (Hazan et al., 2019). This can be address either by learning a non-Markov policy, or by learning a Markov policy that imitates the desired state distribution rather than the demonstrations.

**Traditional Causal Transformer**

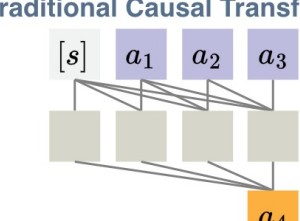

**Chunking Causal Transformer**

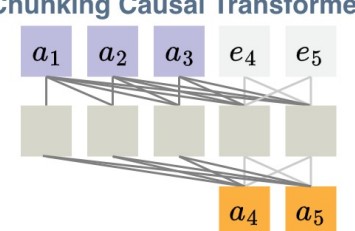

Figure A1: **Causal Transformer versus Chunking Causal Transformer.** Causal transformer prepends the input sequence with a "start" token $[s]$ and modifies the token embedding with causal attention so that the last token $a_3$ becomes the next token $a_4$. Chunking Causal Transformer (CCT) appends the input sequence with a chunk of empty tokens, for example, $e_4, e_5$. CCT modifies the token embedding with causal attention for the known tokens $a_1, a_2, a_3$ and bidirectional attention for the empty tokens $e_4, e_5$. The empty tokens $e_4, e_5$ become the next tokens $a_4, a_5$. CCT can predict multiple next tokens by having more empty tokens.

Table A2: Hyperparameters used in our experiments on ALOHA

| Hyperparameter | Value |
| --- | --- |
| *Model* | |
| number of layers | 4 |
| embedding size | 512 |
| mlp size | 2048 |
| number of latents (gmm) | 1 |
| backbone | RN50 |
| *Action Sequence* | |
| horizon (joints) | 100 |
| horizon (waypoints) | 10 |
| number of action steps | 100 |
| chunk size (joints) | 100 |
| chunk size (waypoints) | 1 |
| *Train & Eval* | |
| observation | RGB $1 \times 480 \times 640 \times 3$ |
| control frequency | 50 |
| maximum evaluation steps | 400 |
| train steps | 100000 |
| eval frequency | 10000 |
| batch size | 8 |
| learning rate | 1.00e-5 |
| learning rate scheduler | none |
| optimizer | AdamW |

Table A3: Hyperparameters used in our experiments on RLBench.

| Hyperparameter | Value |
| --- | --- |
| *Model* | |
| number of layers | 8 |
| embedding size | 128 |
| mlp size | 512 |
| backbone | MVT |
| *Action Sequence* | |
| chunk size | mix of 2 and 1 |
| *Train & Eval* | |
| observation | RGBD $4 \times 128 \times 128 \times 4$ |
| maximum evaluation steps | 25 |
| train epochs | 80000 |
| eval frequency | 10000 |
| batch size | 96 |
| learning rate | 1.25e-5 |
| learning rate scheduler | cosine |
| optimizer | LAMB |

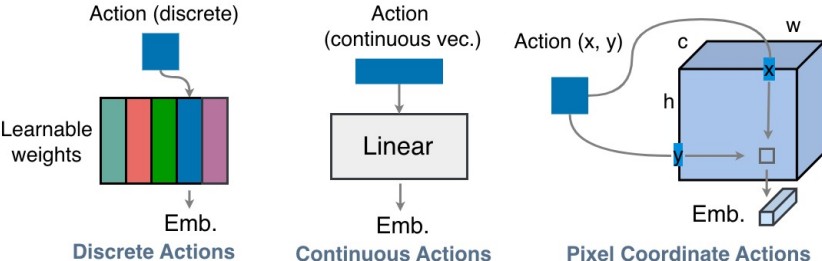

Figure A2: **Embeddings for Discrete, Continuous, and Pixel-coordinate Actions.** Discrete actions are embedded by a simple table lookup on a weight matrix. Continuous actions are embedded with a linear layer. Pixel-coordinate actions are embedded by retrieving the point-wise features at the coordinates on the visual feature maps.

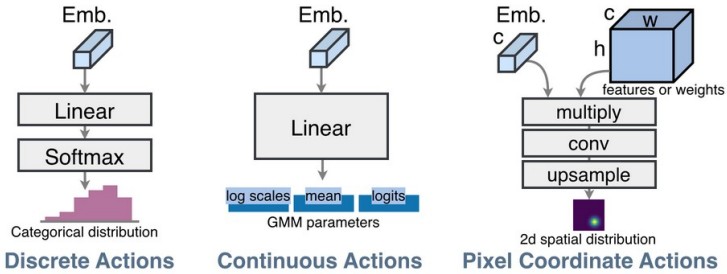

Figure A3: **Decoders for Discrete, Continuous, and Pixel-coordinate Actions.** For discrete actions, we decode the action embeddings into a categorical distribution with a linear layer followed by a softmax operation. For continuous actions, we decode the embeddings into the parameters of a Gaussian mixture distribution with a linear layer. For the pixel-coordinate actions, we multiply the embedding with a visual feature map or a weight tensor, and convert the result into a 2-d heatmap.

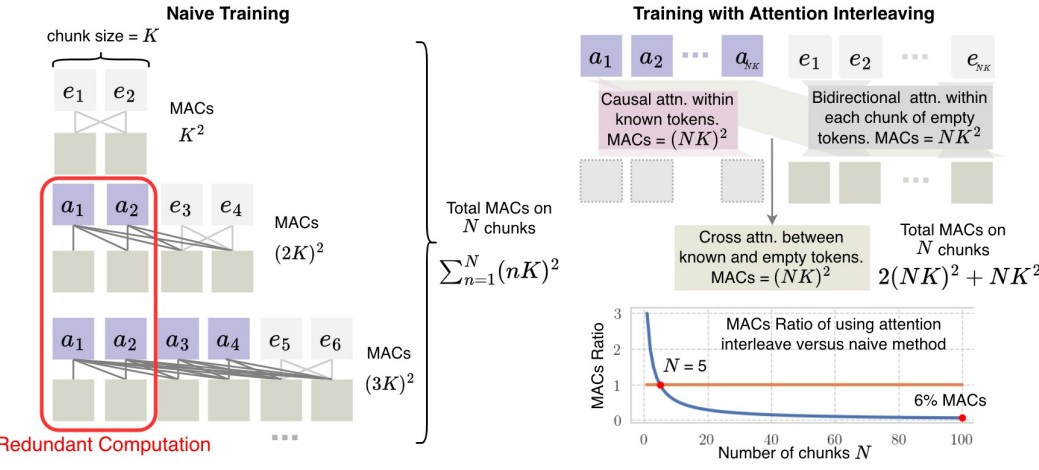

Figure A4: **Naive Training versus Training with Attention Interleaving.** The left figure demonstrates that the causal attention within $a_1, a_2$ is computed twice, when inputs are $a_1, a_2, e_3, e_4$ and $a_1, a_2, a_3, a_4, e_5, e_6$. This redundancy can be reduced by precomputing the causal attention of all known tokens and caching the results. In doing so, the MACs are reduced from $\sum_{n=1}^{N}(nK)^2$ to $2(NK)^2 + NK^2$, where $N$, $K$ are chunk number and chunk size. For simplicity, we count the MACs as the number of attention entries. In addition to the reduced MACs, we find that having a single forward pass for all tokens yields a much cleaner training procedure, a benefit that is not quantified by the raw number of multiply-accumulate operations.

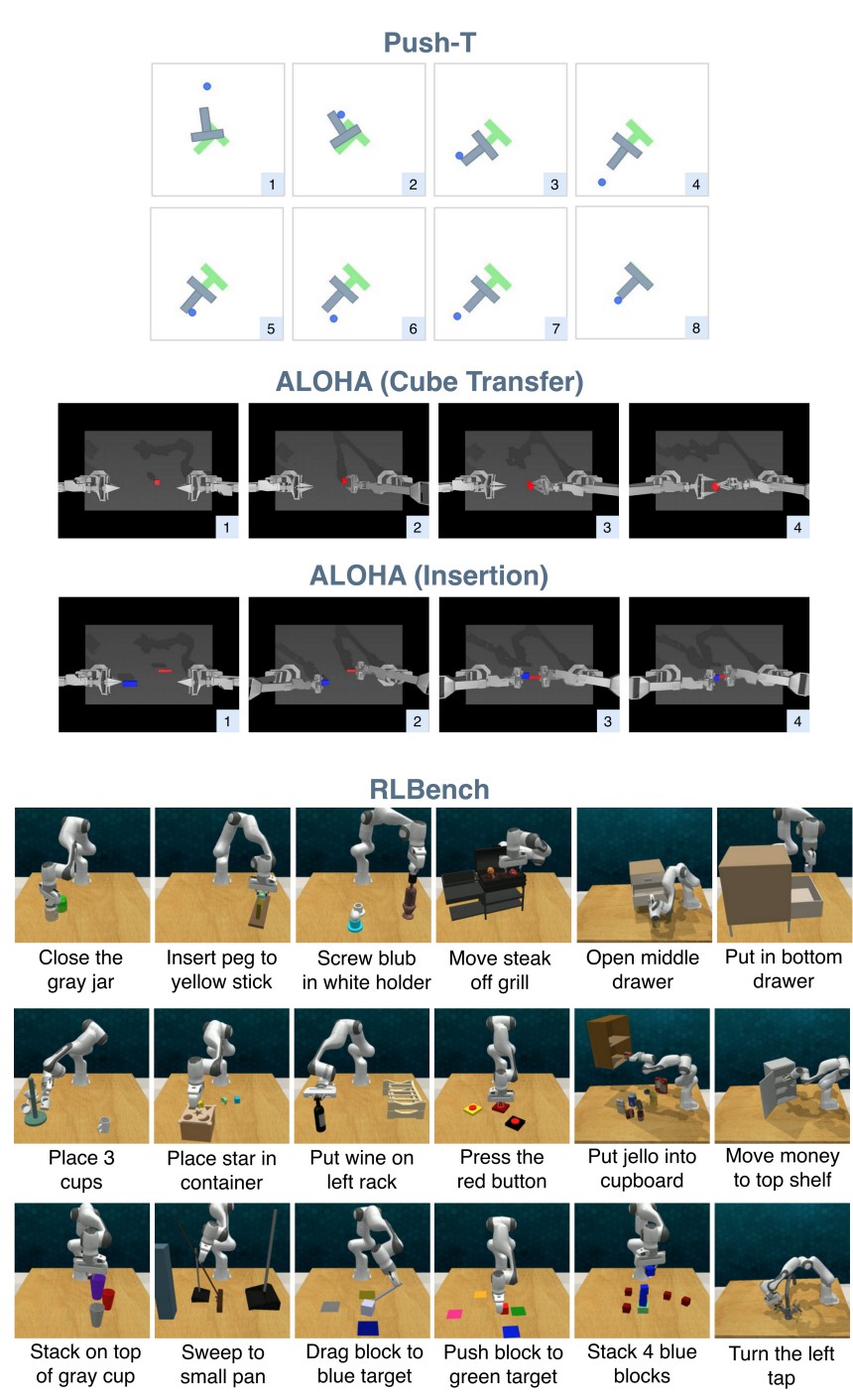

Figure A5: **Demonstrations of all tasks in Push-T, ALOHA, and RLBench.** We provide visualizations of key frames from a single episode of Push-T and ALOHA, with the frame order indicated at the bottom right. For RLBench, we visualize one language variant for each task. RLBench features over 100 task variants specified through natural language commands (James et al., 2020), such as `"open [pos] drawer"` where `pos` is selected from `top, middle, bottom`, and `"stack [num] [color] blocks"`, where `num` ranges from `2, 3, 4`, and `color` is chosen from a palette of 20 colors.

Table A4: **Performance on RLBench.** We report the success rate for each task, and measure the average success rate and rank across all tasks. ARP$^+$ shares the same network definition with ARP but has more layers. The MACs / parameter sizes of RVT-2, ARP, ARP$^+$ are 72.1M/57.1G, 71.9M/56.2G, and 73.8M/57.4G, respectively. ARP performs comparably or outperforms RVT-2 on all tasks. Notably, ARP$^+$ achieves a 97.3% success rate on the challenging peg insertion task.

| Method | Avg. Success | Avg. Rank | Close Jar | Drag Stick | Insert Peg | Meat off Grill | Open Drawer | Place Cups | Place Wine | Push Buttons |
|---|---|---|---|---|---|---|---|---|---|---|
| RVT2 | 77.0 | 2.22 | **100.0** | 90.7 | 30.7 | 96.0 | 89.3 | 18.7 | 89.3 | 88.0 |
| ARP (Ours) | 81.2 | 1.89 | **100.0** | 86.7 | 42.7 | 96.0 | **90.7** | **49.3** | 92.0 | **100.0** |
| ARP$^+$ (Ours) | **86.0** | **1.61** | 96.0 | **100.0** | **97.3** | **97.3** | 88.0 | **49.3** | **94.7** | **100.0** |

| | Put in Cupboard | Put in Drawer | Put in Safe | Screw Bulb | Slide Block | Sort Shape | Stack Blocks | Stack Cups | Sweep to Dustpan | Turn Tap |
|---|---|---|---|---|---|---|---|---|---|---|
| RVT2 | 60.0 | **100.0** | **98.7** | 90.7 | 88.0 | 42.7 | 61.3 | 60.0 | 93.3 | 89.3 |
| ARP (Ours) | 69.3 | **100.0** | 97.3 | 85.3 | **98.7** | 34.7 | 52.0 | 76.0 | 90.7 | **100.0** |
| ARP$^+$ (Ours) | **74.7** | 98.7 | 86.7 | 89.3 | 93.3 | **46.7** | **62.7** | **81.3** | **98.7** | 93.3 |

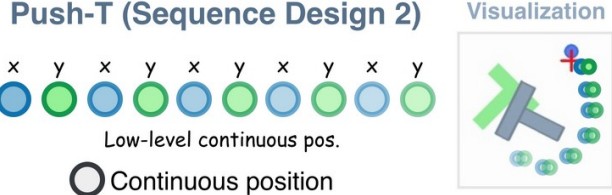

Figure A6: **Flattened Action Sequence for Push-T.** Based on the action sequence in Figure 3, we remove the high-level waypoints and flatten the 2D coordinates into a single vector. For example, a trajectory of $(x_1, y_1), (x_2, y_2), (x_3, y_3)$ is transformed into vector $(x_1, y_1, x_2, y_2, x_3, y_3)$. The policy is trained to predict first the x-coordinate of the initial point, then the y-coordinate, followed by the x- and y-coordinates of subsequent points.

