# OpenReview forum: "Autoregressive Action Sequence Learning for Robotic Manipulation"
_ICLR.cc/2025/Conference — ICLR 2025 Conference Withdrawn Submission_

### Official Review · Reviewer_pk4C · 2024-10-26

**Soundness:** 2
**Presentation:** 3
**Contribution:** 1
**Rating:** 5
**Confidence:** 3

**Summary:**

This paper proposes an autoregressive architecture for robotic manipulation tasks, which changes single-token prediction to multi-token prediction and constructs an autoregressive reasoning pattern by modifying the token embedding with causal attention for past actions. The effectiveness of the proposed structure is verified by comparing it with previously advanced methods in three simulated environments. Experiments on real robots also confirm the superiority of the model. Additionally, this paper conducts a detailed ablation study to thoroughly validate the performance improvements brought about by the autoregressive design.

**Strengths:**

The paper is well-written and easy to follow. The article includes ample details of experiments and methods to aid in understanding and reproducibility.

The paper conducts numerous analytical experiments, qualitative experiments, and discussions to attempt to analyze the performance changes brought about by each module in the method, as well as to explain the working mechanisms of the autoregressive pattern under specific conditions (e.g., prediction with human guidance).

From the experimental demo on real robot, it can be observed that in some cases, the model did not initially execute the task to the expected state (for example, the gripper did not reach the designated position). But in subsequent judgments, the model was able to correct its previous mistakes (by moving a small distance in the correct direction) and successfully complete the task.

**Weaknesses:**

The main weaknesses of this paper lie in its limited novelty and the unclear motivation and contributions compared to related work.

Firstly, transforming the output from a single token to multiple tokens cannot be considered a significant contribution, as many existing models are also capable of multi-step output. And the effectiveness of action chunking has been explored in many cases such as Diffusion Policy and ACT.

Secondly, while the paper proposes an auto-regressive design, it does not provide sufficient comparison and discussion with existing related models in the robotics field. For instance, in my understanding, the PLEX [1] and GR-1 [2] models also use the action outputs from previous steps as inputs to assist in the output of subsequent actions. They also employ a transformer structure similar to GPT. The distinctions and connections between the model proposed in this paper and these models have not been discussed.

[1] Thomas, G., Cheng, C. A., Loynd, R., Frujeri, F. V., Vineet, V., Jalobeanu, M., & Kolobov, A. (2023, December). Plex: Making the most of the available data for robotic manipulation pretraining. In Conference on Robot Learning (pp. 2624-2641). PMLR.

[2] Wu, H., Jing, Y., Cheang, C., Chen, G., Xu, J., Li, X., ... & Kong, T. (2023). Unleashing large-scale video generative pre-training for visual robot manipulation. arXiv preprint arXiv:2312.13139.

**Questions:**

The author should provide a more detailed exposition or validation of the necessity of using the auto-regressive design proposed in this paper for robot manipulation, and engage in discussions with related works.

---

### Official Review · Reviewer_NN6C · 2024-11-03

**Soundness:** 2
**Presentation:** 3
**Contribution:** 2
**Rating:** 3
**Confidence:** 4

**Summary:**

1.  Proposes a CCT model for a multi-token prediction scheme.
2.  CCT is trained with teacher-forcing using a novel attention interleaving method.
3.  The CCT-based ARP model shows improved performance in push-T, ALOHA, and RLBench.

**Strengths:**

1. Multi-token prediction improves inference speed and accuracy.
2. Attention interleaving allows the model to be trained with teacher forcing.
3. A real-world environment is also used for evaluating the model.

**Weaknesses:**

1. Multi-token prediction for action chunks has been explored in many previous works, such as their cited Diffusion Policy and MDT (Multimodal Diffusion Transformer: Learning Versatile Behavior from Multimodal Goals). I acknowledge that they are diffusion-based policies, but it shows that the multi-token prediction scheme has been explored before and thus is not very novel.
2. Teacher-forcing is a common training method for sequence-to-sequence models. The fact that ARP needs special attention interleaving to enable teacher-forcing might be due to the model's complexity. For example, Gato can be easily trained with teacher-forcing without any special attention interleaving.
3. The experiment section is limited in the following ways:
    - Push-T is a simple environment, with 2-dim action, a single object, and a fixed goal position. Therefore, the results of push-T in figure-7 might not be sufficient to show the effect of chunk size.
    -  I saw ALOHA has more than the mentioned two tasks, can the author explain why only two tasks are chosen?
    - Missing ablation study: the benefit of the proposed methods, multi-token prediction and teacher forcing, is not shown
    - Only a few baselines are included, a more diverse set of baselines can better show the effectiveness of ARP. For example, Gato, VIMA, DT, etc.
4. Figure 5 might be clearer if shown as a table.

**Questions:**

1. What is the reason for "alternates between self-attention within the input embeddings and cross-attention with the vision features"? Does this provide any advantage over self-attention-only models (e.g., Gato) or cross-attention-only models (e.g., VIMA)?
2. Because ARP uses action chunking, is temporal smoothing/temporal ensemble also used during inference, following previous works like ACT and MT-ACT?
3. Since RLBench has 18 tasks, why not provide more detailed results/analysis for individual tasks?

---

### Official Review · Reviewer_Luy2 · 2024-11-04

**Soundness:** 2
**Presentation:** 2
**Contribution:** 2
**Rating:** 5
**Confidence:** 4

**Summary:**

This paper introduces a transformer-based policy for robotic tasks, combining (i) multi-token predictions and (ii) autoregressive generation. To speed up training, it proposes attention interleaving, which precomputes intermediate attention results for reuse across operations. The approach is evaluated on three simulated benchmarks and one real-world task.

**Strengths:**

- The proposed approach to action sequence prediction looks new and interesting. While recent studies have pursued multi-token prediction or autoregressive generation separately, the potential benefits of their combination are underexplored and worth studying.
- The analysis of design choices, such as chunk size, autoregression, and hierarchical structure, is extensive and clarifies the core contributions in general.

**Weaknesses:**

- The technical insights are somewhat limited. In particular, Table 1 shows autoregression outperforms multi-token prediction; why is multi-token prediction still used? Further insight into their combined benefit could strengthen the contribution.
- The chunk size analysis in Section 4.2 is difficult to interpret. The results in scenarios 2,3,4 are contrary to recent literature [1,2], where longer chunks are often favored over no chunks for temporal dependencies. Additional details, e.g., default high-/low-level chunk size, and action horizons, could make this analysis clearer.
- Experimental baselines seem limited. For a new architecture involving action chunks and hierarchical structures, comparisons to recent architectures of action chunks (e.g., VQ-BET[3]) or hierarchical structures (HDP[4]) would be more convincing.
- The real-world experiment does not look rigorous, with only 10 evaluation trials and no comparison to recent baselines.
Either expanding the evaluation or removing real-world experiments would make the paper more sound.


[1] Diffusion Policy: Visuomotor Policy Learning via Action Diffusion \
[2] Bidirectional Decoding: Improving Action Chunking via Closed-Loop Resampling \
[3] VQ-BeT: Behavior Generation with Latent Actions \
[4] Hierarchical Diffusion Policy for Kinematics-Aware Multi-Task Robotic Manipulation

**Questions:**

- Given that both token count and autoregression steps influence prediction length, how are these parameters chosen in practice?
- In the chunk size analysis, what is the action horizon (number of executed steps before replanning), and what are the high- and low-level chunk sizes in Figures 7.2 and 7.4?

---

### Official Review · Reviewer_uLNp · 2024-11-06

**Soundness:** 2
**Presentation:** 3
**Contribution:** 2
**Rating:** 3
**Confidence:** 4

**Summary:**

The paper proposes an autoregressive architecture for policy learning. The model architecture, called chunking causal transformer, predicts several future actions in a single forward pass, rather than predicting one action at the time. This is achieved by feeding empty tokens corresponding to future actions as input. The use of input empty tokens poses a challenge for efficient training with parallel prediction with causal attention which the paper proposes to address using attention interleaving which performs attention caching. The approach is quantitatively evaluated on 3 simulated robot environments and tested on a real robot task as well.

**Strengths:**

- Developing better architectures for policy learning can have a large impact in practice
- The paper describes the proposed architecture clearly and with detailed diagrams
- The proposed approach outperforms the baselines in tested settings

**Weaknesses:**

- The novelty of the proposed architecture is limited. The overall autoregressive architecture is fairly standard. The main architectural change is the use of empty tokens for multi-step prediction.
- The empirical results are limited to relatively simple settings in simulation. It is a bit hard to draw conclusions about robotic manipulation performance based on these experiments alone.
- Related work can be improved. It would be good to focus more on architectural changes which are the main focus of the paper. It would also be good to relate the use of empty tokens for multi-step prediction to masked architectures [1].

[1] "Robot learning with sensorimotor pre-training.", Radosavovic et al., Conference on Robot Learning. PMLR, 2023.

**Questions:**

Please see above.

---

### Note · Authors · 2024-11-12

I have read and agree with the venue's withdrawal policy on behalf of myself and my co-authors.